# Impact of major depression on cardiovascular outcomes for individuals with hypertension: prospective survival analysis in UK Biobank

Nicholas Graham [1], Joey Ward,[1] Daniel Mackay,[2] J P Pell,[2] Jonathan Cavanagh,[2] Sandosh Padmanabhan,[3] Daniel J Smith [4]

[1]Gartnavel Royal Hopsital, University of Glasgow Institute of Health and Wellbeing, Glasgow, UK
[2]1 Lilybank Gardens, Institute of Health and Wellbeing, University of Glasgow, Glasgow, UK
[3]Institute of Cardiovascular and Medical Sciences, British Heart Foundation Glasgow Cardiovascular Research Centre, University of Glasgow, Glasgow, UK
[4]Institute of Health and Wellbeing, University of Glasgow, Glasgow, UK

**Correspondence to**
Dr Nicholas Graham;
nicholas.graham@glasgow.ac.uk

## ABSTRACT

**Objectives** To assess whether a history of major depressive disorder (MDD) in middle-aged individuals with hypertension influences first-onset cardiovascular disease outcomes.

**Design** Prospective cohort survival analysis using Cox proportional hazards regression with a median follow-up of 63 months (702 902 person-years). Four mutually exclusive groups were compared: hypertension only (n=56 035), MDD only (n=15 098), comorbid hypertension plus MDD (n=12 929) and an unaffected (no hypertension, no MDD) comparison group (n=50 798).

**Setting** UK Biobank.

**Participants** UK Biobank participants without cardiovascular disease aged 39–70 who completed psychiatric questions relating International Classification of Diseases-10 Revision (ICD-10) diagnostic criteria on a touchscreen questionnaire at baseline interview in 2006–2010 (n=134 860).

**Primary and secondary outcome measures** First-onset adverse cardiovascular outcomes leading to hospital admission or death (ICD-10 codes I20–I259, I60–69 and G45–G46), adjusted in a stepwise manner for sociodemographic, health and lifestyle features. Secondary analyses were performed looking specifically at stroke outcomes (ICD-10 codes I60–69 and G45–G46) and in gender-separated models.

**Results** Relative to controls, adjusted HRs for adverse cardiovascular outcomes were increased for the hypertension only group (HR 1.36, 95% CI 1.22 to 1.52) and were higher still for the comorbid hypertension plus MDD group (HR 1.66, 95% CI 1.45 to 1.9). HRs for the comorbid hypertension plus MDD group were significantly raised compared with hypertension alone (HR 1.22, 95% CI 1.1 to 1.35). Interaction measured using relative excess risk due to interaction (RERI) and likelihood ratios (LRs) were identified at baseline (RERI 0.563, 95% CI 0.189 to 0.938; LR p=0.0116) but not maintained during the follow-up.

**Limitations** Possible selection bias in UK Biobank and inability to assess for levels of medication adherence.

**Conclusions** Comorbid hypertension and MDD conferred greater hazard than hypertension alone for adverse cardiovascular outcomes, although evidence of interaction between hypertension and MDD was inconsistent over time. Future cardiovascular risk prediction tools may

## Strengths and limitations of this study

▶ Methodological advantages over previous studies, including a very large sample size, adjustment for a more comprehensive range of confounders, and the inclusion of non-fatal adverse cardiovascular events from hospital admission data and death registry data.

▶ Definition of prior major depressive disorder (MDD) history was based on International Classification of Diseases-10 Revision diagnostic criteria (rather than a score on a symptoms questionnaire) and our composite definition of hypertension incorporated history, current medication and objective blood pressure measurements.

▶ Although analyses were adjusted for a broad range of baseline factors (such as smoking status, body mass index, psychotropic medication use and diabetes), we were unable to account for how these factors may have changed over the course of follow-up, or assess adherence to cardiovascular medications.

▶ Trained nurses interviewed UK Biobank participants, but the self-report nature of some of these data may represent a limitation.

▶ UK Biobank may have issues with respect to selection biases. For example, individuals with more severe MDD may have been less likely to volunteer.

benefit from the inclusion of questions about prior history of depressive disorders.

## INTRODUCTION

By 2030, major depressive disorder (MDD) and cardiovascular disease (CVD) will be the two leading causes of disability worldwide.[1] MDD is associated with CVD and worse long-term outcomes.[2] To date, survival analysis in comorbid hypertension and MDD have focused on all-cause death[3–5] CV death[5] or incorporated individuals with previous CVD,[3–6] and have suggested a possible additive interaction between hypertension and MDD on survival.[5 6] MDD is well known to

worsen post-CV event survival.[6 7] The contribution on survival to first-onset CVD is less clear when MDD is stratified by hypertension and no prior study has assessed comorbid MDD and hypertension on first episode CVD. Within this study, we look specifically at first-onset events, irrespective of whether they lead to death or not.

Hypertension is extremely common (affecting 1 billion people worldwide)[8] and is responsible for 50% of all CVD.[9] It is commonly comorbid with MDD,[10 11] with recent meta-analysis showing 27% of individuals with hypertension having MDD[12] and population-based studies showing a hypertension prevalence of 21% in those with MDD.[11] A biological link has been found by genome-wide association studies, showing calcium-channel genes, important in blood pressure (BP) control and hypertension,[13] also act to increase risk for MDD[14 15] and bipolar disorder (BD).[16 17] The sympathetic nervous system (SNS), renin–angiotensin system, the immune system and the cortisol stress response system are all also implicated in both conditions.[18] Medication management of both conditions are also thought to impact one another with side effects of psychotropic medications including raised BP and vice versa,[19–21] although there is contrary evidence suggesting either medication or MDD may in actual fact be protective of hypertension.[20 22]

Here, we make use of prospective data from the UK Biobank cohort[23] to test the hypothesis that a lifetime history of MDD in individuals with hypertension impacts adversely on first-episode CV events. We also assess whether MDD exacerbates the effects of hypertension as a risk factor for CVD.

## METHODS
### Study design
This was a population cohort study using data from UK Biobank. Four mutually exclusive groups (hypertension only, MDD only, hypertension plus MDD and a comparison group) were compared for adverse CVD and stroke outcomes.

### Sample description
UK Biobank is a large cohort of 502 655 participants recruited between April 2007 and July 2010 from 21 assessment centres located across Great Britain.[23] Participants aged 39–70 were invited to take participate if registered with the National Health Service (NHS) and lived within a reasonable distance of an assessment centre. At baseline assessment participants completed a series of detailed assessments relating to lifestyle and medical history on touchscreen questionnaire and have a range of physical health measurements, including body mass index (BMI) and BP taken by a nurse.

During the last 2 years of recruitment, questions relating to mood disorder features were added to the baseline assessment schedule questionnaire. From the 172 729 participants asked these questions, 134 860 provided sufficient responses to be included in our analysis.

Participants were excluded based on the following *a priori* criteria: a history of BD (n=1831) or schizophrenia (n=262); where there were insufficient data provided by participants to clearly rule out MDD (n=25 520) or hypertension (n=1080); and where there were coding errors for date and/or time of death (n=4). These exclusions were based on self-report (individuals who listed schizophrenia or BD from a list of pre-existing medical conditions), or criteria for BD as per Smith *et al*,[24] or where they responded 'don't know' or 'prefer not to answer' to questions or data were missing that would limit our ability to exclude the presence of hypertension or MDD. Participants were further excluded from the adverse CVD outcome if they had a record of CVD prior to recruitment (self-reported angina, myocardial infarction (MI) or stroke based on specific questions, or previous hospital admission for angina, MI or stroke) (n=9172). For the stroke outcome, this exclusion was limited to a record of stroke prior to baseline assessment (self-report or previous hospital admission for stroke) (n=2280).

### Classification of hypertension and MDD
Participants were defined as having hypertension if either: (1) mean BP at baseline was greater than clinically defined criteria over two measurements (systolic BP greater than or equal to 140 mm Hg or diastolic BP greater than or equal to 90 mm Hg. Where only one reading was available this was used (n=1571)) or (2) self-reported 'hypertension diagnosed by a doctor' plus self-report of currently taking antihypertensive medication. This composite classification was used to ensure that undiagnosed hypertensive participants were not misclassified and is in line with similar epidemiological studies.[5 25 26] The requirement for antihypertensive use in the context of a history of hypertension was incorporated to limit those on beta-blockers for anxiety. According to these criteria, n=68 964 participants (51.1% of the sample) had hypertension for the adverse CV outcomes analysis and n=73 671 participants (52% of the sample) had hypertension in the stroke outcome analysis.

A history of lifetime MDD was defined according to the criteria for mood disorders developed by Smith *et al*[24 27] and has been used in further papers[27–31] (n=28 027 adverse CV outcomes; n=29 528 stroke outcomes). Participants were classified as having a history of MDD if they reported at least one episode, which comprised depression and/or irritability, with a duration of at least 2 weeks, plus had consulted with either a general practitioner or psychiatrist for mental ill health. This classification followed the structured diagnostic approach within the International Classification of Diseases (ICD)[24] and is described in more detail within the online supplementary content.

For the adverse CV outcomes, the remainder of the samples with no history of hypertension or MDD (n=50 798) were classified as a comparator group. The three mutually exclusive diagnostic groups for this study were, therefore, hypertension only (n=56 035); MDD only

(n=15 098) and hypertension plus MDD (n=12 929). For the stroke outcomes, the mutually exclusive groups were hypertension only (n=59 724); MDD only (n=15 581) and hypertension plus MDD (n=13 947) and no hypertension—no MDD (n=52 502).

## Outcomes

The primary outcome was defined as a first-episode CV event leading to hospital admission or death, specifically angina, MI or chronic ischaemic heart disease (ICD-10 codes I20–I259), and transient ischaemic attack or stroke (ICD-10 codes I60–69 and G45–G46). A secondary outcome was defined as stroke leading to hospital admission or death (ICD-10 codes I60–69 and G45–G46)[32] due to the strength of relationship hypertension has with this outcome in particular.[9] Admission data were obtained from Hospital Episode Statistics in England, Patient Episode Database for Wales and Scottish Morbidity Records in Scotland. Mortality outcomes were obtained from the NHS Information Centre for England and Wales and from the NHS Central Register for Scotland. Individuals who died from a non-CV cause/stroke were censored at the time of death but not recorded as having an event. Admission data were available for Scottish, English and Welsh participants until 31 August 2014, 31 March 2015 and 28 February 2015, respectively. End of follow-up was classified as these dates unless preceded by date of death or the date of first CV admission.

## Confounding variables

Information on potential confounding factors was available for age, sex, socioeconomic status (Townsend score),[33] self-reported ethnicity, age of leaving full-time education, diabetes, BMI, systolic BP, hypercholesterolaemia, alcohol use, smoking history, sedentary behaviour (number of hours each day spent sitting at a computer, television or driving), physical activity levels[34] and psychotropic medication use. Specific details on these variables are provided in online supplementary content.

## Analyses

Baseline characteristics were compared between groups using $X^2$ tests for categorical variables and Kruskal-Wallis for continuous variables. Confounding variables were assessed for differences in adverse CV outcomes using log rank sums. For the four groups of interest, we assessed associations with adverse CV outcomes using Cox proportional hazard regression and the Efron method for ties.[35] Models were applied in a staged process in line with previous studies[3–5] and reported as unadjusted (model 1), partially adjusted (model 2) and fully adjusted (model 3). Model 2 adjusted for sociodemographic factors (age, sex, Townsend score, age of leaving full-time education and ethnicity) and model 3 additionally adjusted for health and lifestyle factors (diabetes, hypercholesterolaemia, BMI, smoking history, alcohol use, systolic BP, sedentary hours per day, physical activity and psychotropic medication use). The proportionality of hazard assumption

was assessed using Schoenfeld residuals.[36] We compared our fully adjusted models with results from competing risk analyses using the Fine and Gray approach,[37] incorporating non-CV deaths as a competing event for CV events and non-stroke deaths for stroke events. The relative excess risk due to interaction (RERI)[38] was calculated to assess for additivity in the risk. All analyses were performed with Stata statistical software, V.12[39] with the exception of RERI which was calculated using the Microsoft Excel method of Andersson *et al*, which allows for comparison of adjusted outcomes.[40] The presence of multiplicative interaction was calculated using the likelihood ratio test.[41]

Psychotropic medication use was included as a confounding variable because of reports that they may increase risk of mortality[42] but we also conducted a sensitivity analysis which excluded participants who were taking psychotropic medication. Subgroup analyses looking separately at HRs in male and female groups only were also carried out to assess for any gender-specific differences in light of differing rates of depression and adverse CV events in each gender.[24 43]

## Time-varying coefficients

In the context of Schoenfeld residuals showing non-proportionality, models with time-varying coefficients were used. In addition, log(-log) plots were carried out to find the time point at which the proportionality assumption failed. Following this, the data were stratified by time at this time point, effectively creating two separate survival analyses pre and post the failure time point.

## Patient involvement

Although patients were not directly involved with the design of the specific research questions in this study, the hypotheses tested were developed in the context of clinical experience that depression and hypertension may interact to impact on CVD. UK Biobank has an active and ongoing programme of participant involvement: www.ukbiobank.ac.uk/participants/. The outcome measures used were those provided by the UK Biobank data collection protocol, the design of which had input from participants. UK Biobank also has a website and social media streams to disseminate research findings and hosts an annual scientific meeting, which includes cohort participants.

## RESULTS

The final sample for adverse CV outcome included 134 860 participants, followed for a median duration of 63 months (702 901.6 person-years follow-up, mean 62.5 months). In total 3685 (2.73%) participants had a first-episode CV event during the follow-up period (total number of all deaths plus non-fatal CV events=5788) and 910 (0.64%) participants had a first-episode stroke event (total number of all deaths plus non-fatal stroke events=7317).

Table 1 describes the baseline characteristics of the four groups. In general, the hypertension only and comorbid hypertension plus MDD groups were older, had higher BMI and were more likely to have diabetes and hypercholesterolaemia. The MDD only and comorbid hypertension plus MDD groups had a higher proportion of women and were more likely to be current smokers (table 1). Gender-separated descriptive tables are shown in the online supplementary content (online supplementary tables 1 and 2).

The sample for stroke-specific outcomes included 141 754 participants, followed for a median duration of 63 months (735 247.7 person-years follow-up, mean 62.2 months). Table 2 describes the baseline characteristics of the four groups, which display similar characteristics to the adverse CVD outcome groups. Gender-separated descriptive tables are shown in the online supplementary content (online supplementary tables 3 and 4).

## Adverse CV outcomes

Within the main analysis and the female-only subgroup analysis, MDD failed the proportional hazards assumption. Table 3 presents unadjusted and multivariate-adjusted HRs (aHRs) for adverse CV outcomes. In the fully adjusted model, relative to the comparator group, the aHR for adverse CV outcomes was significantly raised for hypertension only (aHR 1. 36, 95% CI 1.22 to 1.52) and higher still for comorbid hypertension plus MDD (aHR 1.66, 95% CI 1.46 to 1.9) but reduced for MDD only (aHR 0.55, 95% CI 0.46 to 0.76). Although the MDD only HR was noted to increase over time as a time-varying coefficient. With the exception of MDD, these findings were robust to sensitivity analysis excluding those on psychotropic medication (sensitivity analysis aHR 1.43, 95% CI 1.27 to 1.62; aHR 1.72, 95% CI 1.49 to 1.999, aHR 0.74, 95% CI 0.52 to 1.06, respectively). Table 4 presents HRs and aHRs for adverse CV outcomes using the hypertension only group as comparator. In the fully adjusted model, relative to hypertension, the aHR for adverse CV outcomes was significantly raised for comorbid hypertension plus MDD (aHR 1.22, 95% CI 1.1 to 1.35, sensitivity analysis aHR 1.20, 95% CI 1.08 to 1.34). An adjusted survival plot is shown in figure 1.

Within the subanalysis, the male-only model showed a significant increase in HR for hypertension (male aHR 1.29, 95% CI 1.13 to 1.47) (online supplementary table 5) and comorbid MDD and hypertension (male aHR 1.47, 95% CI 1.24 to 1.74). However, the difference between comorbid disease and hypertension only was not statistically significant (aHR 1.14, 95% CI 0.995 to 1.3). The female-only subanalysis showed an increase in HR for hypertension (aHR 1.64, 95% CI 1.33 to 2.02) and a greater increase in comorbid MDD and hypertension (aHR 2.18, 95% CI 1.82 to 2.92) online supplementary table 6). The difference between comorbid disease and hypertension only was also statistically significant (aHR 1.33, 95% CI 1.14 to 1.56). Sensitivity analysis supported these findings.

## Stroke outcomes

None of the independent variables for stroke outcome failed the proportionality assumption. Table 5 presents HRs and aHRs for stroke outcomes. In the fully adjusted model, the aHR for stroke was insignificantly raised for hypertension only (aHR 1.21, 95% CI 0.97 to 1.51) and depression only (aHR 1.20, 95% CI 0.89 to 1.63) but significantly raised for comorbid hypertension plus MDD (aHR 1.37, 95% CI 1.04 to 1.79). In the hypertension comparator group, no group was significantly different from hypertension only (table 6). Similar trends were shown in the gender subset analysis but not reaching significance (online supplementary tables 7 and 8). An adjusted survival plot is shown in figure 2. Again, all results were supported by sensitivity analysis excluding those on psychotropic medication.

## Interaction, time-stratified analysis and competing risk analysis

Survival analysis stratified by time is described and included within the online supplementary content (online supplementary tables 9 and 10 and figure 3). There was evidence of both additive and multiplicative interaction between hypertension and MDD at baseline for the overall CV outcome analysis before the 22.5-month time point (additive: RERI 0.563, 95% CI 0.189 to 0.938. Multiplicative: likelihood ratio p=value 0.0116) and the female-only CV endpoint analysis before the 29-month time point (additive: RERI 0.588, 95% CI 0.074 to 1.103. Multiplicative: likelihood ratio p value 0.031). However, after these time points, there was no evidence of interaction on either the additive or multiplicative scale. Online supplementary table 11 shows the full results for this analysis. Competing risk analysis showed no significant difference from the main analyses for CV outcomes or stroke outcomes (tables 7–8).

## DISCUSSION

In this large population cohort of middle-aged adults without CVD (adjusted for a broad range of confounders), individuals with comorbid hypertension and MDD were at increased risk of CVD when compared with those with hypertension alone, MDD alone and neither condition. There was some evidence of additive and multiplicative interaction between hypertension and MDD at baseline, but not throughout follow-up and only within the female subgroup. Such a finding may suggest a causal interaction between MDD and hypertension in females only, but suggests that this may be limited over time leading to a suspected further interaction with a gender-specific unmeasured confounder. Differences between comorbid disease and either disease alone or no disease were more marked in females. For stroke outcomes, comorbid depression and hypertension was the only group that showed significantly increased HRs.

## Previous research

Our findings expand on previous research from UK Biobank looking at CVD in those with BD and MDD.[27]

**Table 1** Baseline characteristics for adverse cardiovascular outcomes

| | Comparator group n=50 798 | Hypertension only n=56 035 | MDD only n=15 098 | Hypertension plus MDD n=12 929 |
|---|---|---|---|---|
| Median age (range)* | 54 (47–61) | 61 (55–65) | 53 (46–60) | 60 (53–64) |
| Females, N (%) | 2928 (57.54) | 25 893 (46.21) | 10 929 (72.39) | 7676 (59.37) |
| Ethnicity, N (%) | | | | |
| White | 46 147 (90.84) | 51 249 (91.46) | 14 247 (94.36) | 12 272 (94.92) |
| Asian/Asian British | 1771 (3.49) | 1696 (3.03) | 261 (1.73) | 179 (1.38) |
| Black/ Black British | 1323 (2.6) | 1769 (3.16) | 219 (1.45) | 222 (1.72) |
| Median Townsend score (range)* | −1.89 (−3.45 to 0.54) | −2.07 (−3.51 to 0.39) | −1.64 (−3.3 to 0.93) | −1.84 (−3.42 to 0.76) |
| Age at leaving full-time education, N (%) | | | | |
| <16 | 5916 (11.65) | 12 085 (21.57) | 1725 (11.43) | 2607 (20.16) |
| 16 | 10 265 (20.21) | 11 827 (21.11) | 3178 (21.05) | 2732 (21.13) |
| >16 | 34 090 (67.1) | 31 480 (56.18) | 10 090 (66.83) | 7503 (58.03) |
| Total physical activity in metabolic | 3.97 (1.68–8.03) | 3.79 (1.51–8.03) | 3.89 (1.66–8) | 3.68 (1.49–7.95) |
| Sedentary time in hours, median (range)* | 4 (3–6) | 4.5 (3.5–6) | 4.5 (3–6) | 5 (3.5–6) |
| Diabetes, N (%) | 1268 (2.5) | 3777 (6.74) | 380 (2.52) | 929 (7.19) |
| Hypercholesterolaemia, N (%) | 3011 (5.93) | 9210 (16.44) | 893 (5.91) | 2211 (17.1) |
| Systolic BP in mm Hg, median (range)* | 125.5 (118–132) | 149.5 (142–159.5) | 124 (116–131) | 147.5 (140.5–157) |
| Body mass index kg/m², N (%) | | | | |
| <18.5 | 389 (0.77) | 142 (0.25) | 103 (0.68) | 34 (0.26) |
| 18.5–25 | 22 549 (44.39) | 13 678 (24.41) | 6251 (41.4) | 2874 (22.23) |
| 25–30 | 20 410 (40.18) | 25 216 (45) | 5936 (39.32) | 5389 (41.68) |
| >30 | 7450 (14.67) | 16 999 (30.34) | 2808 (18.6) | 4632 (35.83) |
| Smoking status, N (%) | | | | |
| Never smoked | 30 626 (60.29) | 31 503 (56.22) | 7864 (52.09) | 6454 (49.92) |
| Previously smoked | 15 056 (29.64) | 20 140 (35.94) | 5118 (33.9) | 5065 (39.18) |
| Current smoker | 4970 (9.78) | 4199 (7.49) | 2093 (13.8) | 1381 (10.68) |
| Alcohol frequency, N (%) | | | | |
| Daily or almost daily | 9450 (18.6) | 12 970 (23.15) | 2736 (18.12) | 2881 (22.28) |
| Three or four times a week | 12 175 (23.97) | 13 033 (23.26) | 3253 (21.55) | 2837 (21.94) |
| Once or twice a week | 13 644 (26.86) | 13 889 (24.79) | 3880 (25.7) | 2916 (22.55) |
| One to three times a month | 6052 (11.91) | 5588 (9.97) | 2058 (13.63) | 1512 (11.69) |
| Special occasions only | 5534 (10.89) | 6330 (11.3) | 1904 (12.61) | 1729 (13.37) |

Continued

**Table 1** Continued

| | Comparator group n=50 798 | Hypertension only n=56 035 | MDD only n=15 098 | Hypertension plus MDD n=12 929 |
|---|---|---|---|---|
| Never | 3924 (7.72) | 4199 (7.49) | 1262 (8.36) | 1048 (8.11) |
| Psychotropic medication, N (%) | 1341 (2.64) | 1795 (3.2) | 2844 (18.84) | 2522 (19.51) |

All data presented as N (%) and has X² p<0.001 except * which are median values (IQR) and have a Kruskal-Wallis p of 0.0001. Data presented as MET-hours (metabolic equivalents: hours per week spent doing exercise adjusted for multiples of basal metabolic rate in accordance with international physical activity questionnaire). Townsend score is an area based measure based on census statistics. It is a calculation based on the number of: households without a car, overcrowded households, households not owner occupied and unemployment.
BP, blood pressure; MDD, major depressive disorder.

It was found that there were significantly increased odds of having 'any CVD' (fully adjusted OR 1.15, 95% CI 1.12 to 1.19) or hypertension (fully adjusted OR 1.15, 95% CI 1.13 to 1.18) if depressed, with even higher odds for stroke (fully adjusted OR 1.26, 95% CI 1.13 to 1.40). There are distinct differences between our current paper and the previous publication. Follow-up data within UK Biobank have been released to allow meaningful prospective studies be conducted. Thus, the current paper has the benefits of using hospital records and death certification for outcomes, rather than self-reported data. We are also able to make inferences about the direction of effect regarding MDD and CVD and assess the influence of hypertension and MDD over time, both in isolation and when comorbid, and assess for statistical interaction to inform on whether there may be a biological interaction.

Other survival analyses in hypertension/MDD comorbidity have focused primarily on mortality outcomes. In the National Health and Nutrition Epidemiologic Follow-up Study in the [31]USA and the Taiwanese Survey of Health and Living Status,[32] individuals with self-reported hypertension plus depressive symptoms (compared with a reference group with neither) had increased all-cause mortality (aHR 1.39, 95% CI 1.14 to 1.69, aHR 1.54, 95% CI 1.29 to 1.83, respectively)[3 4] with the former also showing increased CVD-specific mortality (aHR 1.59, 95% CI 1.08 to 2.34).[4] Similarly, Hamer et al[5] reported a prospective analysis of common mental disorder on mortality outcomes in individuals with hypertension versus those without hypertension in participants from the Health Survey for England and the Scottish Health Survey (1994–2004), finding that risk of CVD death was highest in the group with comorbid disease.

### Strengths

These observations are broadly consistent with our results but our study has a number of methodological advantages, including a very large sample size, adjustment of analyses for a more comprehensive range of confounders and a focus on first-episode non-fatal and fatal adverse CV events. We also used a definition of prior MDD history which was based on diagnostic criteria within ICD-10 (rather than a threshold score on a depressive symptoms or general well-being scale) and our composite definition of hypertension incorporated history, baseline medication and BP measurements. Lifetime MDD is thought to be under-reported in the literature. However, using current symptom scores may reduce power and precision because a smaller number of respondents would be identified as having an episode of MDD.[44] Given that we are assessing outcomes for which risk accumulates over a lifetime, we felt that a primary focus on lifetime episodes was appropriate. We believe our lifetime definition to be better suited as it offers a view depression and depressive symptoms over the course of a lifespan as opposed the past week. Also, within our current study, we were able to exclude those with previous self-declared or hospital

**Table 2**  Baseline characteristics for stroke outcomes

| | Comparator group | Hypertension only | MDD only | Hypertension plus MDD |
| --- | --- | --- | --- | --- |
| | n=52 502 | n=59 724 | n=15 581 | n=13 947 |
| Median age (range)* | 54 (47–61) | 61 (55–65) | 54 (47–61) | 60 (53–64) |
| Females, N (%) | 29 684 (56.54) | 26 937 (45.1) | 11 143 (71.52) | 8090 (58.01) |
| Ethnicity, N (%) | | | | |
| White | 47 697 (90.85) | 54 578 (91.38) | 14 697 (94.33) | 13 212 (94.73) |
| Asian/Asian British | 1857 (3.54) | 1889 (3.16) | 280 (1.8) | 209 (1.5) |
| Black/ Black British | 1355 (2.58) | 1854 (3.1) | 223 (1.43) | 246 (1.76) |
| Median Townsend score (range)* | −1.89 (−3.45 to 0.55) | −2.04 (−3.49 to 0.44) | −1.56 (−3.28 to 1.15) | −1.74 (−3.4 to 0.93) |
| Age at leaving full-time education, N (%) | | | | |
| <16 | 6446 (12.28) | 13 396 (22.43) | 1884 (12.09) | 2945 (21.12) |
| 16 | 10 590 (20.17) | 12 507 (20.94) | 3270 (20.99) | 2953 (21.17) |
| >16 | 34 914 (66.5) | 33 114 (55.45) | 10 317 (66.22) | 7947 (56.98) |
| Total physical activity in metabolic | 3.96 (1.67–8.02) | 3.75 (1.5–8) | 4.13 (1.67–8.36) | 3.66 (1.45–7.83) |
| Sedentary time in hours, median (range)* | 4 (3–6) | 5 (3.5–6) | 5 (3.5–6.5) | 5 (4–7) |
| Diabetes, N (%) | 1454 (2.77) | 4502 (7.54) | 449 (2.88) | 1163 (8.34) |
| Hypercholesterolaemia, N (%) | 3592 (6.84) | 10 768 (18.03) | 1049 (6.73) | 2620 (18.79) |
| Systolic BP in mm Hg, median (range)* | 125.5 (118–132) | 149.5 (142–159.5) | 127 (120.5–133) | 147.5 (140.5–156.5) |
| Body mass index (kg/m$^2$), N (%) | | | | |
| <18.5 | 395 (0.75) | 151 (0.25) | 104 (0.67) | 38 (0.27) |
| 18.5–25 | 22 967 (43.75) | 14 242 (23.85) | 6374 (40.91) | 3017 (21.63) |
| 25–30 | 21 185 (40.35) | 26 817 (44.9) | 6149 (39.46) | 5769 (41.36) |
| >30 | 7953 (15.15) | 18 514 (31) | 2954 (18.96) | 5123 (36.73) |
| Smoking status, N (%) | | | | |
| Never smoked | 31 318 (59.65) | 32 982 (55.22) | 8052 (51.68) | 6834 (49) |
| Previously smoked | 15 851 (30.19) | 22 019 (36.87) | 5340 (34.27) | 5560 (39.87) |
| Current smoker | 5170 (9.85) | 4501 (7.54) | 2163 (13.88) | 1519 (10.89) |
| Alcohol frequency, N (%) | | | | |
| Daily or almost daily | 9760 (18.59) | 13 751 (23.02) | 2817 (18.08) | 3085 (22.12) |
| Three or four times a week | 12 563 (23.93) | 13 827 (23.15) | 3335 (21.4) | 3020 (21.65) |
| Once or twice a week | 14 089 (26.84) | 14 719 (24.65) | 3993 (25.63) | 3125 (22.41) |
| One to three times a month | 6220 (11.85) | 5971 (10) | 2122 (13.62) | 1627 (11.67) |
| Special occasions only | 5744 (10.94) | 6794 (11.38) | 1978 (12.69) | 1885 (13.52) |
| Never | 4102 (7.81) | 4630 (7.75) | 1330 (8.54 | 1199 (8.6) |
| Psychotropic medication, N (%) | 1408 (2.68) | 1996 (3.34) | 2976 (19.1) | 2778 (19.92) |

All data presented as N (%) and has X$^2$ p <0.001 except * which are median values (IQR) and have a Kruskal-Wallis p of 0.0001. Data presented as MET-hours (metabolic equivalents: hours per week spent doing exercise adjusted for multiples of basal metabolic rate in accordance with International Physical Activity Questionnaire). Townsend score is an area based measure based on census statistics. It is a calculation based on the number of: households without a car, overcrowded households, households not owner occupied and unemployment.
BP, blood pressure; MDD, major depressive disorder.

**Table 3** Risk of adverse cardiovascular event by clinical group: unadjusted, partially adjusted and fully adjusted models

| | Unadjusted | | | Model 1—sociodemographic* | | | Model 2—model 1+health/lifestyle† | | |
|---|---|---|---|---|---|---|---|---|---|
| | HR | 95% CI | P value | aHR | 95% CI | P value | aHR | 95% CI | P value |
| Group | | | | | | | | | |
| No hypertension—no MDD | 1(ref) | | | 1(ref) | | | 1(ref) | | |
| Hypertension only | 2.60 | (2.39 to 2.82) | $3.31\times10^{-113}$ | 1.72 | (1.57 to 1.88) | $1.99\times10^{-33}$ | 1.36 | (1.22 to 1.52) | $2.92\times10^{-8}$ |
| MDD only | 0.69 | (0.51 to 0.94) | 0.02 | 0.82 | (0.6 to 1.13) | 0.23 | 0.75 | (0.54 to 1.04) | 0.08 |
| Hypertension and MDD | 2.84 | (2.55 to 3.17) | $6.31\times10^{-77}$ | 2.27 | (2.02 to 2.55) | $2.75\times10^{-44}$ | 1.66 | (1.45 to 1.9) | $7.48\times10^{-14}$ |
| Time-varying variables | | | | | | | | | |
| MDD only | 1.01 | (1.004 to 1.02) | $2.38\times10^{-3}$ | 1.01 | (1.004 to 1.02) | $3.19\times10^{-3}$ | 1.01 | (1.004 to 1.02) | $3.03\times10^{-3}$ |

*Adjusted for sociodemographic factors (age, sex, Townsend score, age of leaving full-time education and ethnicity.
†Additionally adjusted for history of diabetes, history of hypercholesterolaemia, BMI, smoking history, alcohol use, systolic blood pressure, sedentary hours per day, physical activity and psychotropic medication use.
aHR, adjusted HR; BMI, body mass index; MDD, major depressive disorder.

**Table 4** Risk of adverse cardiovascular event by clinical group: unadjusted, partially adjusted and fully adjusted models with hypertension as the comparator

| | Unadjusted | | | Model 1—sociodemographic* | | | Model 2—model 1+health/lifestyle† | | |
|---|---|---|---|---|---|---|---|---|---|
| | HR | 95% CI | P value | aHR | 95% CI | P value | aHR | 95% CI | P value |
| Group | | | | | | | | | |
| Hypertension only | 1(ref) | | | 1(ref) | | | 1(ref) | | |
| No hypertension—no MDD | 0.38 | (0.35 to 0.42) | $3.31\times10^{-113}$ | 0.58 | (0.53 to 0.63) | $1.99\times10^{-33}$ | 0.73 | (0.66 to 0.82) | $2.92\times10^{-8}$ |
| MDD only | 0.27 | (0.2 to 0.36) | $1.14\times10^{-17}$ | 0.48 | (0.35 to 0.66) | $4.91\times10^{-6}$ | 0.55 | (0.4 to 0.76) | $3.23\times10^{-4}$ |
| Hypertension and MDD | 1.09 | (0.996 to 1.2) | 0.06 | 1.32 | (1.2 to 1.46) | $3.07\times10^{-8}$ | 1.22 | (1.1 to 1.35) | $1.30\times10^{-4}$ |
| Time-varying variables | | | | | | | | | |
| MDD only | 1.01 | (1.004 to 1.02) | 0.002 | 1.01 | (1.004 to 1.02) | $3.19\times10^{-3}$ | 1.01 | (1.004 to 1.02) | $3.03\times10^{-3}$ |

*Adjusted for sociodemographic factors (age, sex, Townsend score, age of leaving full-time education and ethnicity.
†Additionally adjusted for history of diabetes, history of hypercholesterolaemia, BMI, smoking history, alcohol use, systolic blood pressure, sedentary hours per day, physical activity and psychotropic medication use.
aHR, adjusted HR; BMI, body mass index; MDD, major depressive disorder.

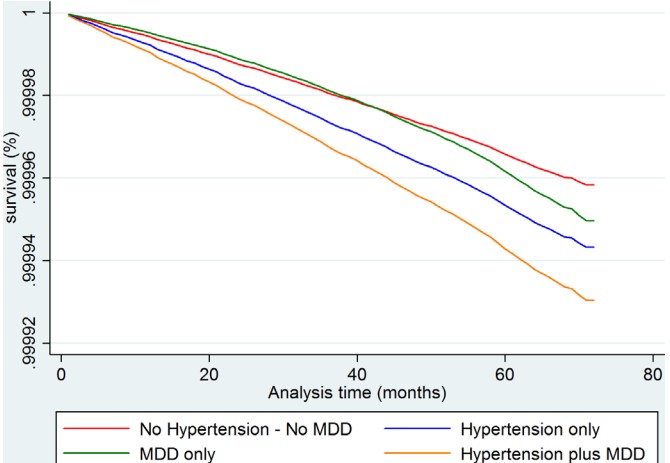

**Figure 1** Adjusted survival analysis graph for adverse cardiovascular outcome showing greatest hazard for the comorbid group. MDD appears protective compared with the comparator group initially, however, shows increased hazard after 41 months. Analysis adjusted for age, sex, Townsend score, age of leaving full-time education and ethnicity, history of diabetes, history of hypercholesterolaemia, BMI, smoking history, alcohol use, systolic blood pressure, sedentary hours per day, physical activity and psychotropic medication use. BMI, body mass index; MDD, major depressive disorder.

admission CVD, as previous studies show depression may result from CVD[45 46] and worsen prognosis.[46]

### Limitations

However, some limitations are acknowledged. Recruitment criteria for UK Biobank may lead to selection bias. Specifically, age restrictions may lead to under-representation of early-onset hypertension and those with more severe MDD may be less inclined to attend for assessment. We also acknowledge limitations with our classifications of MDD and hypertension, which were primarily self-report rather than formal diagnostic assessments. Although we have excluded prior CV events where possible, the MDD plus hypertension subtype may capture older individuals with a degree of vascular depression, which has an established association with raised BP.[47] In addition, although we adjust for a host of risk factors at baseline such as smoking status, BMI and psychotropic medication, we are limited by the lack of follow-up data, which could show change and modification of said risk factors over time. Similarly, we were unable to assess for medication adherence and transitions from one investigatory group to another. Participants who are aware of or had sought treatment for MDD may also have complicated our findings, however, our sensitivity analysis excluded those using pharmaceutical treatments and was in keeping with our main findings. Such modifications could explain the non-proportional nature of the depression group, which may in itself be a predictor of poor medication adherence.[48] Although adherence to medication was not formally assessed, the number and duration of antihypertensive medications used in the hypertension plus MDD group was the same as for the hypertension only group

**Table 5** Risk of stroke event by clinical group: unadjusted, partially adjusted and fully adjusted models

| Group | Unadjusted | | | Model 1—sociodemographic* | | | Model 2—model 1+health/lifestyle† | | |
|---|---|---|---|---|---|---|---|---|---|
| | HR | 95% CI | P value | aHR | 95% CI | P value | aHR | 95% CI | P value |
| No hypertension- no MDD | 1(ref) | | | 1(ref) | | | 1(ref) | | |
| Hypertension only | 2.55 | (2.16 to 3.02) | $3.84 \times 10^{-28}$ | 1.64 | (1.38 to 1.96) | $3.35 \times 10^{-8}$ | 1.21 | (0.97 to 1.51) | 0.09 |
| MDD only | 1.14 | (0.86 to 1.52) | 0.37 | 1.37 | (1.02 to 1.84) | 0.037 | 1.20 | (0.89 to 1.63) | 0.24 |
| Hypertension and MDD | 2.67 | (2.13 to 3.34) | $9.79 \times 10^{-18}$ | 2.05 | (1.63 to 2.58) | $1.08 \times 10^{-9}$ | 1.37 | (1.04 to 1.79) | 0.02 |

*Adjusted for sociodemographic factors (age, sex, Townsend score, age of leaving full-time education and ethnicity.
†Additionally adjusted for history of diabetes, history of hypercholesterolaemia, BMI, smoking history, alcohol use, systolic blood pressure, sedentary hours per day, physical activity and psychotropic medication use.
aHR, adjusted HR; BMI, body mass index; MDD, major depressive disorder.

**Table 6** Risk of stroke event by clinical group: unadjusted, partially adjusted and fully adjusted models with hypertension as the comparator

| Group | Unadjusted | | | Model 1—sociodemographic* | | | Model 2—model 1+health/lifestyle† | | |
|---|---|---|---|---|---|---|---|---|---|
| | HR | 95% CI | P value | aHR | 95% CI | P value | aHR | 95% CI | P value |
| Hypertension only | 1(ref) | | | 1(ref) | | | 1(ref) | | |
| No hypertension—no MDD | 0.39 | (0.33 to 0.46) | $3.84 \times 10^{-28}$ | 0.61 | (0.51 to 0.73) | $3.35 \times 10^{-8}$ | 0.82 | (0.66 to 1.03) | 0.09 |
| MDD only | 0.45 | (0.34 to 0.58) | $1.43 \times 10^{-9}$ | 0.83 | (0.63 to 1.1) | 0.19 | 0.99 | (0.73 to 1.35) | 0.95 |
| Hypertension and MDD | 1.05 | (0.86 to 1.27) | 0.64 | 1.25 | (1.03 to 1.52) | 0.03 | 1.13 | (0.92 to 1.39) | 0.26 |

*Adjusted for sociodemographic factors (age, sex, Townsend score, age of leaving full-time education and ethnicity.
†Additionally adjusted for history of diabetes, history of hypercholesterolaemia, BMI, smoking history, alcohol use, systolic blood pressure, sedentary hours per day, physical activity and psychotropic medication use.
aHR, adjusted HR; BMI, body mass index; MDD, major depressive disorder.

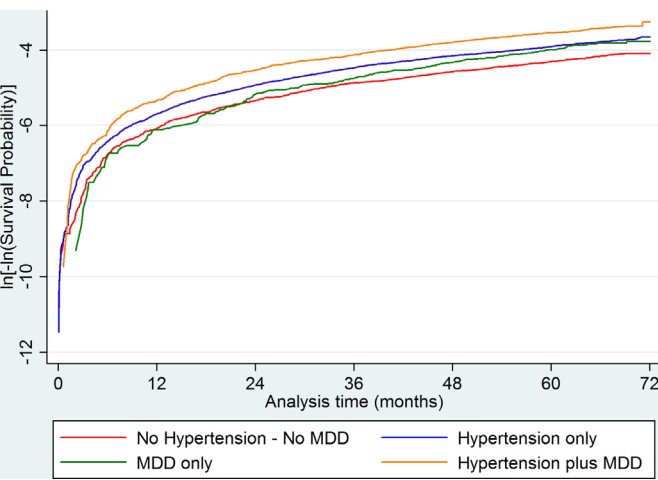

**Figure 2** Adjusted survival analysis graph for stroke outcomes showing significantly increased hazard for comorbid hypertension and MDD, with similar insignificant increased hazard trends for hypertension only and MDD only. analysis adjusted for age, sex, Townsend score, age of leaving full-time education and ethnicity, history of diabetes, history of hypercholesterolaemia, BMI, smoking history, alcohol use, systolic blood pressure, sedentary hours per day, physical activity and psychotropic medication use. BMI, body mass index; MDD, major depressive disorder.

(online supplementary content, online supplementary table 12). As such, worse outcomes in the MDD plus hypertension group are not explained by less intensive antihypertensive treatment at baseline. The endpoints used for stroke and CV events also require to be further validated, however, are in line with previous epidemiological studies[5] and have been suggested in previous papers in UK Biobank.[32] CV endpoints have not, to our

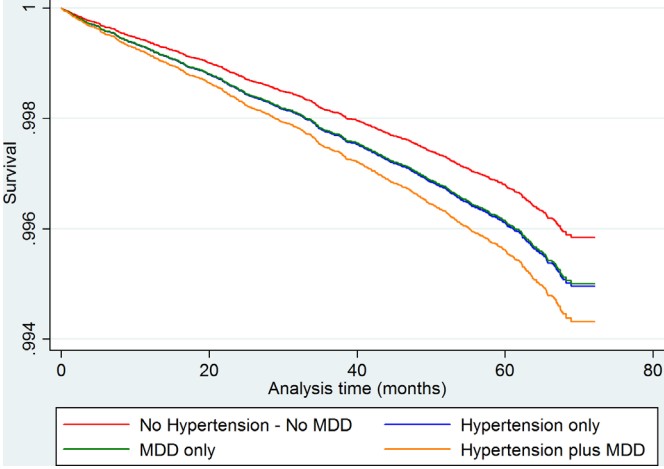

**Figure 3** Log (-log) plot showing non-proportionality of MDD only survival over time. Paths between the comparator group and the MDD group cross at the 22.5 months mark. Analysis adjusted for age, sex, Townsend score, age of leaving full-time education and ethnicity, history of diabetes, history of hypercholesterolaemia, BMI, smoking history, alcohol use, systolic blood pressure, sedentary hours per day, physical activity and psychotropic medication use. BMI, body mass index; MDD, major depressive disorder.

**Table 7** Fully adjusted HR compared with results from competing risks analysis for cardiovascular endpoints

|  | Fully adjusted non-competing risks analysis | | | Fully adjusted competing risks model | | |
|---|---|---|---|---|---|---|
|  | aHR | 95% CI | P value | aHR | 95% CI | P value |
| Group |  |  |  |  |  |  |
| No hypertension—no MDD | 1(ref) |  |  | 1(ref) |  |  |
| Hypertension only | 1.36 | (1.22 to 1.52) | $2.92 \times 10^{-8}$ | 1.37 | (1.22 to 1.53) | $4 \times 10^{-8}$ |
| MDD only | 0.75 | (0.54 to 1.04) | 0.08 | 0.76 | (0.55 to 1.03) | 0.08 |
| Hypertension and MDD | 1.66 | (1.45 to 1.9) | $7.48 \times 10^{-14}$ | 1.67 | (1.45 to 1.91) | $2.2 \times 10^{-13}$ |
| Tvc |  |  |  |  |  |  |
| MDD only | 1.01 | (1.004 to 1.02) | $3.03 \times 10^{-3}$ | 1.01 | (1.004 to 1.02) | 0.003 |

Adjusted for age, sex, Townsend score, age of leaving full-time education and ethnicity, history of diabetes, history of hypercholesterolaemia, BMI, smoking history, alcohol use, systolic blood pressure, sedentary hours per day, physical activity and psychotropic medication use.
aHR, adjusted HR; BMI, body mass index; MDD, major depressive disorder.

knowledge, been validated within UK Biobank, however, we do not feel that this will bias the results towards any particular group. The amelioration of the aHR suggests other covariates contribute considerably to the risk. This is important in the context of increased rates of diabetes, hypercholesterolaemia and obesity along with lower socioeconomic status in the hypertension only and comorbid groups and as such we may be seeing the summation of CV risk factors. Finally, the overall recruitment rate to UK Biobank was low (at around 6%); however, the large final cohort size, the depth and diversity of phenotype data collected at baseline and the wide sociodemographic representation of participants all make our findings highly relevant to UK primary care settings. While UK Biobank participants cannot be used to provide representative disease prevalence and incidence rates, valid assessment of exposure–disease relationships are nonetheless widely generalisable and do not require participants to be representative of the UK population at large,[49] although findings will not be generalisable to other countries.

### Possible mechanisms
Our finding that a history of MDD, in the context of a current diagnosis of hypertension increased the risk of first-episode CVD, is complicated by the time-varying risk that MDD conveys to CVD. Subsample analysis show this time-varying aspect is gender specific to females. Within our sample, the MDD group has a slightly reduced BP compared with comparators. Previously, reduced BP has been put forth as being causative of MDD and therefore reducing CVD risk,[20] but findings from longitudinal studies are inconsistent with regard to direction of effect.[50 51] Potential menopausal effects are tempting explanations. Common factors for BP and mood such as neuropeptide Y[52 53] may also influence CV outcomes. Neuropeptide Y is linked tovasoconstriction and down-regulated by oestrogen[54] . Neuropeptide Y and oestrogen may represent a biologically plausible interaction between MDD and hypertension, however, this would require investigation.

Personality factors may also play a role. MDD correlates highly with neuroticism which, although inconsistent, may be protective of CVD.[56] Conscientiousness traits may lead to better outcomes[57] and it is possible that this trait has been selected for within UK Biobank. Despite this early reduced risk, due to the time-varying nature of MDD, MDD has increased risk in the latter aspects of the time-stratified analyses for the full and female-only analyses (online supplementary tables 9 and 10). The findings from our study in this context suggest that MDDs role as a risk factor for CVD and its relationship with BP may be much more complex than initially thought, in particular within female populations; however, further investigation is clearly needed.

**Table 8** Fully adjusted HR compared with results from competing risks analysis for stroke endpoints

|  | Fully adjusted non-competing risks analysis | | | Fully adjusted competing risks model | | |
|---|---|---|---|---|---|---|
| Group | aHR | 95% CI | P value | aHR | 95% CI | P value |
| No hypertension—no MDD | 1(ref) |  |  | 1(ref) |  |  |
| Hypertension only | 1.21 | (0.97 to 1.51) | 0.09 | 1.21 | (0.96 to 1.52) | 0.1 |
| MDD only | 1.20 | (0.89 to 1.63) | 0.24 | 1.20 | (0.88 to 1.64) | 0.25 |
| Hypertension and MDD | 1.37 | (1.04 to 1.79) | 0.02 | 1.36 | (1.03 to 1.8) | 0.031 |

Adjusted for age, sex, Townsend score, age of leaving full-time education and ethnicity, history of diabetes, history of hypercholesterolaemia, BMI, smoking history, alcohol use, systolic blood pressure, sedentary hours per day, physical activity and psychotropic medication use.
aHR, adjusted HR; BMI, body mass index; MDD, major depressive disorder.

We can see in the hypertension-only baseline models that comorbid hypertension and depression convey a significantly greater risk than hypertension alone. Individuals with either hypertension or depression may have increased sympathetic stimulation that is increased further in comorbid states leading to worse outcomes.[58]

## CONCLUSIONS

Overall, our findings may have important implications for routine clinical practice, particularly within primary care settings and further demonstrate the complex relationship between depression and hypertension. Although evidence of an interaction is inconsistent, we found that comorbid hypertension and depression conferred greater hazard than hypertension alone for adverse CV outcomes. This significant finding remained after adjustment for factors such as BMI, smoking status and diabetes and was robust to sensitivity analysis excluding those on psychotropic medication. One implication is that clinicians should be more aware of the negative long-term impact on CVD outcomes caused by a history of MDD in the context of hypertension, particularly within females. Although this work awaits replication and testing in other cohorts and settings, further work in this field may suggest that future iterations of CVD risk prediction tools, such as ASSIGN,[59] would benefit from the addition of a question on whether individuals have a history of MDD, to facilitate more intensive support to prevent CVD.[60]

**Acknowledgements** We are grateful to all participants of the UK Biobank cohort. UK Biobank was established by the Wellcome Trust, the Medical Research Council, Department of Health, Scottish Government and the Northwest Regional Development Agency. It has also had funding from the Welsh Assembly Government and the British Heart Foundation. UK Biobank is hosted by the University of Manchester and supported by the National Health Service (NHS). NG is supported by the Aitchison Family Clinical Research Fellowship at the University of Glasgow and DJS is supported by a Lister Institute Prize Fellowship. JC is supported by the Sackler Trust and the Wellcome Trust. Part funded by the Medical Research Council Mental Health Data Pathfinder Award (grant reference MC_PC_17217).

**Contributors** NG, JW, JPP, JC, DJS, SP and DM contributed to study design and writing of the manuscript. JPP and DM contributed to data acquisition. NG conducted data processing and statistical analyses.

**Funding** The authors have not declared a specific grant for this research from any funding agency in the public, commercial or not-for-profit sectors.

**Competing interests** None declared.

**Patient consent for publication** Not required.

**Ethics approval** This study has been conducted using UK Biobank data. UK Biobank has received ethics approval from the UK Biobank Research Ethics Committee (ref. 11/NW/0382).

**Provenance and peer review** Not commissioned; externally peer reviewed.

**Data availability statement** Data may be obtained from a third party and are not publicly available.

**ORCID iDs**
Nicholas Graham http://orcid.org/0000-0002-5332-0783
Daniel J Smith http://orcid.org/0000-0002-2267-1951

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
