## [Reviewer comments · BMJ Open]

ARTICLE DETAILS

TITLE (PROVISIONAL)	Impact of major depression on cardiovascular outcomes for individuals with hypertension: prospective survival analysis in UK Biobank.
AUTHORS	Graham, Nicholas; Ward, Joey; Mackay, Daniel; Pell, J. P.; Cavanagh, Jonathan; Padmanabhan, Sandosh; Smith, Daniel

VERSION 1 – REVIEW

REVIEWER	Nathalie Moise Columbia University Medical Center, USA
REVIEW RETURNED	12-Jul-2018

GENERAL COMMENTS	Overall, this is an interesting and well written study. Strengths include the fact that this is a large, known, multicenter cohort with presumably rigorously collected data (UK biobank), with rigorous hypertension classification and focus on CVD outcomes and events. It adds to the debate on the relationship between hypertension and MDD each of which alone confers increased risk of CVD events, but is limited by the poorly described MDD classification, selection bias, and lack of clear description of gaps in the literature/novelty. In addition, the description of the mechanisms would benefit from being rewritten. Abstract It is difficult to ascertain how this is at all novel in the abstract. https://www.ncbi.nlm.nih.gov/pubmed/27040355 The authors should specify if this is depressive symptoms or MDD measured by psychiatric interview The phrase additive interaction is not clear. Limitation should be noted in the abstract Introduction The introduction focuses on the limited attention paid to the relationship between depression and hypertension (e.g., authors mention genetic links between the 2 disease states). Their introduction does little to support their premise and approach for focusing on the additive effect of MDD and hypertension on CVD, however. Relatedly, the introduction lacks an adequate rationale for how their study is novel in comparison to other studies on MDD and hypertension. Prior MDD and CVD studies have controlled for hypertension and hypertension studies may also have controlled for depression. The author should lay out the rationale for needing to know the additive effect on CVD (has this never been done?).
--

	The authors should specify/quantify the degree to which depression and hypertension are comorbid. https://www.ncbi.nlm.nih.gov/pubmed/26252317 There is a complex relationship between depression and hypertension (e.g., antidepressants and antihypertensives effects http://hyper.ahajournals.org/content/53/4/63) and it's not clear from the introduction that the authors understand this complexity and how this informs their approach for using mutually exclusive groups. https://www.ncbi.nlm.nih.gov/pubmed/25084798 Methods The author should specify if participants were recruited from the community, primary care or were they recently hospitalized cohort (what are assessment centers?). It would be helpful to specify how the assessments were made at baseline (nurse, self-report, online?) The authors should specify what questions related to mood disorders were added (line 101). Were these self-report, psychiatric evaluations, hospital records of mood disorders? Were those with a history of psychiatric illness excluded based on self-reported data or ICD10? The authors should provide an example of what they mean by insufficient data to rule out MDD (line 106). The only exclusion criteria that is clearly defined is the history of CVD. For strokes, the authors state that record of only stroke prior to admission. For classification of hypertension, did all patients have their blood pressure measured at baseline? While the classification of hypertension is rigorous (authors ensured that those with undiagnosed hypertension were included and in line with epidemiologic studies), this only further highlights how limited the classification of MDD is. They should specify the criteria for MDD (was a validated screening tool used). Why not include those with at least 1 episode or those seen by a general practitioner or psychiatrist for mental health. The latter also opens the door to including those with other mental health disorders (e.g., anxiety). Did this include an lifetime episode or a recent episode? In the strengths/limitations section, the authors report that the MDD classification was based on ICD10. That is not described here. If the authors did use ICD10 codes, this is subject to selection bias (in that those with ICD 10 codes were likely more likely to have had recognized and potentially even treated depression by prior physicians or even vice versa). It can be surmised perhaps that these participants had more salient symptoms, and excludes those underdiagnosed, leading to misclassification bias as well. This may have influenced the results. Did the authors consider competing risk analyses for nonCVD death (prior literature has established a relationship between HTN and MDD with nonCVD death)
--	---

	Were outcomes adjudicated in any way? How confident are the authors in the accuracy of the coding system in Britain? Confounding variables are not well described. How were each measured (objectively, self- report, validated measures?). Did the authors consider including antihypertension/BP treatment, use of statins and aspirin, and medication adherence. Other comorbidities such as PVD, and other risk factors for CVD should be considered and adjusted. The absolute number of participants with new onset events is quite small. How many events including deaths all together? What was the average follow up time? How many individuals Is there an a priori rationale for subgroup analyses? The authors note that they included time varying covariates (which were included, and how often were they measured) How was missingness accounted for? Results The results would benefit from using aHR to specify adjusted analyses. The authors should remind the readers that sensitivity analyses refer to those not on psychotropic medications Table 1 doesn't describe if there are any significant differences between the groups in terms of characteristics at baseline The reduced risk of MDD on CVD events forces one to surmise that this may be due to the use of a poor measure for MDD. This finding isn't clearly described in the methods, results or abstract but is more apparent in the Table 3. The authors start their paper with a focus on MDD. If their main goal is to assess whether MDD augments the risk of hypertension on CVD, this needs to be clearly laid out. What is the rationale for presenting the results with hypertension as the reference group (why not MDD?) (Table 4) Discussion The authors write differences between hypertension and MDD (256) were marked. They should specify that they mean between comorbid disease and either disease or no disease. Please add references for other survival analyses (line 272) The authors should specify not only how their study differed from prior biobank study, but further emphasize the importance and rationale for adding and assessing comorbid hypertension in the context of this previous work. The authors should specify how their work is novel and differs from prior analyses, particularly NHANES. What do these findings add to the literature?
--	---

	It's not clear how the definition within ICD10 is between than a score on depressive symptoms, which has been the focus of prior literature in this area. While the shared genetic risk factors for MDD and hypertension is interesting, it doesn't quite explain the additive risk on CVD. Prior literature has shown that MDD and hypertension separately confer increased CVD risk even after controlling for multiple risk factors and helping to clarify the novelty/premise in the introduction and throughout will allow the authors to better describe the mechanisms. Overall the possible mechanisms section is unfocused and confusing. Why do the authors believe that even after controlling for multiple risk factors that comorbid hypertension and MDD confers a significant risk for CVD? There is no reference for the cardiovascular side effects of antidepressants (line 337) The authors do little describe why MDD would have a lower risk while hypertension and MDD confer a higher risk compared to no hypertension.
--	---

REVIEWER	Christine Baumgartner Inselspital Bern, Bern University Hospital, University of Bern, Bern, Switzerland.
REVIEW RETURNED	09-Aug-2018

GENERAL COMMENTS	This large, population-based cohort study describes the association of major depression and hypertension with cardiovascular events in UK individuals aged 37-73 years. Compared to persons without hypertension or depression, those with hypertension had an increased hazard of cardiovascular events, while those with comorbid hypertension and depression were had even a higher hazard. These findings contribute to the already quite robust evidence of depression being an independent risk factor for cardiovascular events, which is not an entirely novel finding. As hypertension is also a well known risk factor for cardiovascular events, it is not very surprising that comorbid hypertension and depression is associated with an even higher risk. Abstract: in the conclusion, the authors mention that evidence of an additive interaction is inconsistent. However, they did not present any results regarding additive interaction in the abstract. Page 5 lines 74 and 75: this sentence is somewhat confusing, because the authors don't investigate the association between MDD and hypertension (but the association of hypertension+MDD with cardiovascular events). I would suggest that in the introduction, the authors should more clearly point out the gap in current knowledge that they would like to fill with their study. Page 6, lines 94 and 95: it would be helpful for the reader to get more information about the UK Biobank and its participants. How were the participants selected/recruited, and who were these patients (an age range is indicated in the abstract, but not in the methods)?
--

	Page 6, lines 102-107: a flow chart would be helpful to illustrate the selection of eligible participants. Page 6, line 105: The authors should address the issue of missing data. They excluded 25,520 participants based on insufficient data to rule out MDD. Page 6, line 114-116: In how many of the patients was only one measurement used to diagnose or rule out hypertension? Page 7, line 124, definition of MDD: did this definition of MDD relate to any episode within the entire life? Later in the Discussion (page 14 line 287), the authors mention that the diagnosis of MDD was based on ICD-10 diagnostic criteria – this is not clear from the methods section. Were medical records used to identify corresponding ICD-10 codes? Please clarify. Page 7 lines 119-121 and lines 125-126, and page 8 lines 147-148: I would suggest to move these numbers to the results section. Page 8, line 158: I don't see any results from Chi-squared tests of Kruskal Wallis test in the baseline tables. Also, I'm not sure I understand what the authors mean with "confounding variables were assessed for differences in cardiovascular outcomes (...)", as I don't see any results accordingly. Page 9, lines 165-168: please give the rationale for using two different models. Page 9, line 171: Did the authors test for multiplicative interaction? Page 11 line 216: please mention/repeat which sensitivity analysis the result relates to (I assume exclusion of patients with psychotropic medication). Page 12, line 243: evidence on the association between hypertension and stroke is strong – how do the authors explain that they did not find an association? Discussion: The authors state that they were investigating for an interaction of hypertension and MDD on cardiovascular events (page 9, line 171). No formal test to assess multiplicative interaction is presented, and the authors describe their (inconsistent) findings concerning an additive interaction in the results. It would be helpful if the authors could more clearly describe the objective of assessing an interaction and the interpretation of these results. In the discussion, the authors seem to discuss both depression and hypertension as independent risk factors for cardiovascular events (which is already well known), but not about the significance of a potential interaction. Table 1: please describe the Townsend score in the footnote to the table Formatting:  - Please spell out abbreviations at their first use (e.g. MDD in abstract). - If the authors refer to a supplementary table, the table number should be indicated (e.g. page 10, line 205)
--	---

	 - The very small p-values in tables 3 to 6 are difficult to read. - I would suggest to list the supplementary tables in the order that they are mentioned in the text.
--	---

REVIEWER	Eric Boersma Erasmus MC Rotterdam, Netherlands
REVIEW RETURNED	12-Oct-2018

GENERAL COMMENTS	Nicolas Graham et al. studied the impact of major depression on cardiovascular outcomes for individuals with hypertension, based on the UK Biobank. Aspects of this work that need attention  1. It is not clear why the authors study the relation between MDD and adverse CV outcomes in hypertensive subjects only. Note that there are many other risk CVD risk factors, such as obesity, diabetes, smoking, etc. This reviewer supports the notion of ‘total CVD risk’, rather than singling out one particular risk factor. 2. Comment 1 is even more relevant, since the authors did not find evidence of an interaction between MDD and hypertension with respect to CV outcomes. 3. The authors suggest that CV risk prediction might improve by adding information on MDD (see their conclusion). In terms of risk prediction, what was the performance of the multivariable models that the authors developed (discrimination, calibration)? Can they quantify the improvement of the performance of comparing models with vs. without MDD as risk determinant? 4. The quality of ICD-coding of endpoints needs discussion. 5. Stroke outcomes and non-stroke outcomes are competing events/risks. The authors did not account for this phenomenon.
---

VERSION 1 – AUTHOR RESPONSE

Reviewer #1 statement: “Overall, this is an interesting and well written study. Strengths include the fact that this is a large, known, multicenter cohort with presumably rigorously collected data (UK biobank), with rigorous hypertension classification and focus on CVD outcomes and events. It adds to the debate on the relationship between hypertension and MDD each of which alone confers increased risk of CVD events, but is limited by the poorly described MDD classification, selection bias, and lack of clear description of gaps in the literature/novelty. In addition, the description of the mechanisms would benefit from being rewritten.”

RESPONSE: Thank you – these points are addressed below:

Reviewer #1 statement: “It is difficult to ascertain how this is at all novel in the abstract.

<https://www.ncbi.nlm.nih.gov/pubmed/27040355>

The authors should specify if this is depressive symptoms or MDD measured by psychiatric interview.

The phrase additive interaction is not clear.

Limitation should be noted in the abstract.”

RESPONSE: We have clarified the novelty aspect of this work by including “first-time” cardiovascular events in the abstract. We have also changed the introduction to incorporate this. Further clarification of the assessment is now included: “psychiatric questions relating to ICD-10 diagnostic criteria on

touchscreen questionnaire” And we have defined the phrase ‘additive interaction’ as the relative excess risk due to interaction (RERI), also adding results to demonstrate this. Furthermore, we have now included a limitations section within the abstract.

Reviewer #1 statement: “The introduction focuses on the limited attention paid to the relationship between depression and hypertension (e.g., authors mention genetic links between the 2 disease states). Their introduction does little to support their premise and approach for focusing on the additive effect of MDD and hypertension on CVD, however.

Relatedly, the introduction lacks an adequate rationale for how their study is novel in comparison to other studies on MDD and hypertension. Prior MDD and CVD studies have controlled for hypertension and hypertension studies may also have controlled for depression. The author should lay out the rationale for needing to know the additive effect on CVD (has this never been done?). “ The authors should specify/quantify the degree to which depression and hypertension are comorbid. <https://www.ncbi.nlm.nih.gov/pubmed/26252317>.

There is a complex relationship between depression and hypertension (e.g., antidepressants and antihypertensives effects <http://hyper.ahajournals.org/content/53/4/63>) and it’s not clear from the introduction that the authors understand this complexity and how this informs their approach for using mutually exclusive groups. <https://www.ncbi.nlm.nih.gov/pubmed/25084798>”

RESPONSE: We have now highlighted the novelty of our approach. Specifically, we are looking at the first episode of cardiovascular events. Whilst there is evidence of additive interaction between depression and hypertension in adverse cardiovascular events, such studies focus on or incorporate participants who have had prior cardiovascular events, in which there is good evidence of subsequent depression as well as reduced survival. In addition, we have incorporated information on the comorbidity between depression and hypertension. We thank the reviewer for giving us the opportunity to discuss the complexity of the overlap between mood disorders and raised blood pressure, including medication effects. This has resulted in mutually exclusive groups and a high exclusion rate to allow a cleaner comparison between groups. This is highlighted in the revised text.

Reviewer #1 statement: “The author should specify if participants were recruited from the community, primary care or were they recently hospitalized cohort (what are assessment centers?). It would be helpful to specify how the assessments were made at baseline (nurse, self-report, online?)

The authors should specify what questions related to mood disorders were added (line 101). Were these self-report, psychiatric evaluations, hospital records of mood disorders?

While the classification of hypertension is rigorous (authors ensured that those with undiagnosed hypertension were included and in line with epidemiologic studies), this only further highlights how limited the classification of MDD is. They should specify the criteria for MDD (was a validated screening tool used). Why not include those with at least 1 episode or those seen by a general practitioner or psychiatrist for mental health. The latter also opens the door to including those with other mental health disorders (e.g., anxiety). Did this include an lifetime episode or a recent episode? In the strengths/limitations section, the authors report that the MDD classification was based on ICD10. That is not described here.

If the authors did use ICD10 codes, this is subject to selection bias (in that those with ICD 10 codes were likely more likely to have had recognized and potentially even treated depression by prior physicians or even vice versa). It can be surmised perhaps that these participants had more salient symptoms, and excludes those underdiagnosed, leading to misclassification bias as well. This may have influenced the results.

The authors should provide an example of what they mean by insufficient data to rule out MDD (line 106). The only exclusion criteria that is clearly defined is the history of CVD.

Were those with a history of psychiatric illness excluded based on self-reported data or ICD10?"

RESPONSE: Information on UK Biobank recruitment and data collection have been added to the supplementary file. We have also incorporated the questions used to define the depression group, along with the answers given "don't know /prefer not to answer" that would make it impossible to exclude those with psychiatric conditions. We have clarified the exclusions based on self-report data also.

Reviewer #1 statement: "For strokes, the authors state that record of only stroke prior to admission."

RESPONSE: This has been clarified to state "only stroke prior to baseline assessment"

Reviewer #1 statement: "For classification of hypertension, did all patients have their blood pressure measured at baseline?"

RESPONSE: Almost all. Those that did not have blood pressure measured we excluded from the comparator group and depression only group as we were unable to exclude hypertension. Within the all cardiovascular endpoints, 125 individuals did not have blood pressure measurements in the hypertension only group, and 48 did not have measurements in the comorbid group. Within the stroke only outcomes, 142 individuals did not have blood pressure measurements in the hypertension only group, and 57 did not have measurements in the comorbid group. These individuals were included on the basis that they had declared a history of hypertension and antihypertension medication use.

Reviewer #1 statement: "Did the authors consider competing risk analyses for nonCVD death (prior literature has established a relationship between HTN and MDD with nonCVD death)"

RESPONSE: We have incorporated this suggestion in the revised manuscript. This is presented in a new table comparing the fully adjusted survival analysis with the competing risk analyses.

Reviewer #1 statement: "Were outcomes adjudicated in any way? How confident are the authors in the accuracy of the coding system in Britain?"

RESPONSE: We have expanded the explanation for our choice of outcomes to include evidence and guidance for stroke outcomes, with the caveats highlighted within the paper on stroke outcomes. There is, to our knowledge, no guidance on use of UK Biobank endpoints for cardiac causes of death. Due to the range of potential adverse cardiovascular outcomes, we did not wish to limit the definition of adverse cardiovascular outcomes solely to MI.

Reviewer #1 statement: "Confounding variables are not well described. How were each measured (objectively, self-report, validated measures?). Did the authors consider including antihypertension/BP treatment, use of statins and aspirin, and medication adherence. Other comorbidities such as PVD, and other risk factors for CVD should be considered and adjusted."

RESPONSE: Detailed information on confounding variables is described within the supplementary file, referenced in the main text. There is no facility within UK Biobank to test for adherence; this is noted in our limitations.

We considered the use of the risk factors suggested above but antihypertensive medication was incorporated into the definition of hypertension and there were no specific questions relating to statin, PVD and aspirin. We also felt that, due to the known relationship between psychotropic medication use and cardiovascular events and the fact that one of the main observational groups was depression, it was important to assess this in particular with regards to sensitivity analysis. Specifically, we wished to see if excluding those on psychotropics changed the hazard of those with lifetime depression. Furthermore, as multiple adjustments have already been made we are concerned that inclusion of further covariates may lead to overfitting.

Reviewer #1 statement: “The absolute number of participants with new onset events is quite small. How many events including deaths all together? What was the average follow up time? How many individual.

Is there an a priori rationale for subgroup analyses?

RESPONSE: Detail on the average follow up time and events have now been incorporated. Yes, within MDD there is an increased number of those with depression who are female, in keeping with prior literature. Due to the influence gender has on cardiovascular events, we wished to look at males and females separately. We have updated the text to reflect this.

Reviewer #1 statement: “The authors note that they included time varying covariates (which were included, and how often were they measured)

How was missingness accounted for?”

RESPONSE: Covariates were only measured once at baseline. We appreciate that time varying covariates may suggest repeated measures and as such have changed this to time varying coefficients.

Reviewer #1 statement: “The results would benefit from using aHR to specify adjusted analyses. The authors should remind the readers that sensitivity analyses refer to those not on psychotropic medications

Table 1 doesn't describe if there are any significant differences between the groups in terms of characteristics at baseline

Please add references for other survival analyses (line 272)

The authors write differences between hypertension and MDD (256) were marked. They should specify that they mean between comorbid disease and either disease or no disease.

The authors should specify not only how their study differed from prior biobank study, but further emphasize the importance and rationale for adding and assessing comorbid hypertension in the context of this previous work.

The authors should specify how their work is novel and differs from prior analyses, particularly NHANES. What do these findings add to the literature?

It's not clear how the definition within ICD10 is between than a score on depressive symptoms, which has been the focus of prior literature in this area.

While the shared genetic risk factors for MDD and hypertension is interesting, it doesn't quite explain the additive risk on CVD. Prior literature has shown that MDD and hypertension separately confer increased CVD risk even after controlling for multiple risk factors and helping to clarify the novelty/premise in the introduction and throughout will allow the authors to better describe the mechanisms.

Overall the possible mechanisms section is unfocused and confusing. Why do the authors believe that even after controlling for multiple risk factors that comorbid hypertension and MDD confers a significant risk for CVD?

There is no reference for the cardiovascular side effects of antidepressants (line 337)

The authors do little describe why MDD would have a lower risk while hypertension and MDD confer a higher risk compared to no hypertension.”

RESPONSE: The authors thank the reviewer for these comments and corrections and have sought to amend this. Specifically, we have changed HR to aHR where relevant in the text and tables, and incorporated the results for chi-squared and Kruskal-Wallis tests into the footnotes of table 1 and supplementary tables 1 and 2. We have additionally addressed the points raised about the discussion the rationale for assessing comorbid hypertension and depression, how the work differs for NHANES, and restructured the mechanisms section to relate back to the novelty within the introduction and added explanations for raised comorbid disease risk and lesser MDD risk. Additionally we have incorporated a statement as to why we believe our definition to be better than other depressive scores, being that it looks for depressive symptoms over the course of a lifespan as opposed to the past week.

Reviewer #1 statement: “The reduced risk of MDD on CVD events forces one to surmise that this may be due to the use of a poor measure for MDD. This finding isn’t clearly described in the methods, results or abstract but is more apparent in the Table 3. The authors start their paper with a focus on MDD. If their main goal is to assess whether MDD augments the risk of hypertension on CVD, this needs to be clearly laid out.

What is the rationale for presenting the results with hypertension as the reference group (why not MDD?) (Table 4)”

RESPONSE: This point allows us to re-frame our research appropriately and is tied to the previously raised issue of novelty. Hypertension is a well-known risk factor for first-time cardiovascular events, however, survival analyses for depression are frequently focussed on death outcomes only. There is a well-documented relationship between depressive symptoms and death following cardiovascular events and we wished to look specifically at the first episode of cerebrovascular events in those without prior cardiovascular events. Additionally, we wished to look for an additional risk for those with hypertension and depression. It is of course possible that to induce adverse cardiovascular endpoints, depression does indeed require another cardiovascular illness such as prior MI or hypertension. We have adjusted our text accordingly.

Reviewer #2 statement: “This large, population-based cohort study describes the association of major depression and hypertension with cardiovascular events in UK individuals aged 37-73 years. Compared to persons without hypertension or depression, those with hypertension had an increased hazard of cardiovascular events, while those with comorbid hypertension and depression were had even a higher hazard. These findings contribute to the already quite robust evidence of depression being an independent risk factor for cardiovascular events, which is not an entirely novel finding. As hypertension is also a well known risk factor for cardiovascular events, it is not very surprising that comorbid hypertension and depression is associated with an even higher risk.

Abstract: in the conclusion, the authors mention that evidence of an additive interaction is inconsistent. However, they did not present any results regarding additive interaction in the abstract.”

RESPONSE: We have incorporated these results within the revised abstract.

Reviewer #2 statement: “Page 5 lines 74 and 75: this sentence is somewhat confusing, because the authors don’t investigate the association between MDD and hypertension (but the association of hypertension+MDD with cardiovascular events). I would suggest that in the introduction, the authors should more clearly point out the gap in current knowledge that they would like to fill with their study.”

RESPONSE: We have now explained more clearly the novelty of this paper within the introduction.

Reviewer #2 statement: “Page 6, lines 94 and 95: it would be helpful for the reader to get more information about the UK Biobank and its participants. How were the participants selected/recruited, and who were these patients (an age range is indicated in the abstract, but not in the methods)?”
RESPONSE: We have now adjusted the main text to address these points.

Reviewer #2 statement: “Page 6, lines 102-107: a flow chart would be helpful to illustrate the selection of eligible participants.”
RESPONSE: We considered this but, on balance, feel there are already a relatively large number of tables and figures. Hopefully this is acceptable.

Reviewer #2 statement: “Page 6, line 105: The authors should address the issue of missing data. They excluded 25,520 participants based on insufficient data to rule out MDD. “
RESPONSE: We have adjusted the text to explain more fully the rationale for these exclusions.

Reviewer #2 statement: “Page 6, line 114-116: In how many of the patients was only one measurement used to diagnose or rule out hypertension?”
RESPONSE: We have included this information within the revised manuscript.

Reviewer #2 statement: “Page 7, line 124, definition of MDD: did this definition of MDD relate to any episode within the entire life? Later in the Discussion (page 14 line 287), the authors mention that the diagnosis of MDD was based on ICD-10 diagnostic criteria – this is not clear from the methods section. Were medical records used to identify corresponding ICD-10 codes? Please clarify.
RESPONSE: We have specified the criteria used to define depression within the revised methodology section.

Reviewer #2 statement: “Page 7 lines 119-121 and lines 125-126, and page 8 lines 147-148: I would suggest to move these numbers to the results section.
RESPONSE: We have now changed this.

Reviewer #2 statement: “Page 8, line 158: I don’t see any results from Chi-squared tests of Kruskal Wallis test in the baseline tables. Also, I’m not sure I understand what the authors mean with “confounding variables were assessed for differences in cardiovascular outcomes (...)”, as I don’t see any results accordingly.
RESPONSE: “We have now incorporated these results in the tables and incorporated information on the differences in cardiovascular outcomes.

Reviewer #2 statement: “Page 9, lines 165-168: please give the rationale for using two different models.
RESPONSE: Models have been applied in a staged manner, in line with the approach used in previous studies conducted in this area.

Reviewer #2 statement: “Page 9, line 171: Did the authors test for multiplicative interaction?
RESPONSE: The adjusted analysis is effectively a test for multiplicative interaction where the co-morbid group forms the multiplicative interaction term.

Reviewer #2 statement: “Page 11 line 216: please mention/repeat which sensitivity analysis the result relates to (I assume exclusion of patients with pschotropic medication).
RESPONSE: We have now changed the text to address this.

Reviewer #2 statement: “Page 12, line 243: evidence on the association between hypertension and stroke is strong – how do the authors explain that they did not find an association?”

RESPONSE: We assume the lack of finding a significant association for stroke is due to the low number of events. We have adjusted the revised text accordingly.

Reviewer #2 statement: "Discussion: The authors state that they were investigating for an interaction of hypertension and MDD on cardiovascular events (page 9, line 171). No formal test to assess multiplicative interaction is presented, and the authors describe their (inconsistent) findings concerning an additive interaction in the results. It would be helpful if the authors could more clearly describe the objective of assessing an interaction and the interpretation of these results. In the discussion, the authors seem to discuss both depression and hypertension as independent risk factors for cardiovascular events (which is already well known), but not about the significance of a potential interaction.

RESPONSE: We agree and have revised the discussion section to address this..

Reviewer #2 statement: "Table 1: please describe the Townsend score in the footnote to the table
RESPONSE: This has been added.

Reviewer #2 statement: "Formatting:

- Please spell out abbreviations at their first use (e.g. MDD in abstract).
- If the authors refer to a supplementary table, the table number should be indicated (e.g. page 10, line 205)
- The very small p-values in tables 3 to 6 are difficult to read.
- I would suggest to list the supplementary tables in the order that they are mentioned in the text."

RESPONSE: The authors thank the reviewer for pointing out formatting problems and have adjusted the text accordingly.

Reviewer #3 statement: "

1. It is not clear why the authors study the relation between MDD and adverse CV outcomes in hypertensive subjects only. Note that there are many other risk CVD risk factors, such as obesity, diabetes, smoking, etc. This reviewer supports the notion of 'total CVD risk', rather than singling out one particular risk factor.

2. Comment 1 is even more relevant, since the authors did not find evidence of an interaction between MDD and hypertension with respect to CV outcomes."

RESPONSE: We thank the reviewer for these comments and we have adjusted the introduction to describe more clearly the novelty of this work.

Reviewer #3 statement: "The authors suggest that CV risk prediction might improve by adding information on MDD (see their conclusion). In terms of risk prediction, what was the performance of the multivariable models that the authors developed (discrimination, calibration)? Can they quantify the improvement of the performance of comparing models with vs. without MDD as risk determinant?

RESPONSE: We agree with these suggestions on discrimination and calibration but on balance feel that including these in the revised manuscript is beyond the original scope of this work, although we have softened our conclusions appropriately.

Reviewer #3 statement: "The quality of ICD-coding of endpoints needs discussion.

RESPONSE: We have now incorporated this into the main text.

Reviewer #3 statement: "Stroke outcomes and non-stroke outcomes are competing events/risks. The authors did not account for this phenomenon.

RESPONSE: We have incorporated a new table comparing fully adjusted models to competing risks models.

We thank the reviewers again for their comments and we hope the changes made are acceptable.

VERSION 2 – REVIEW

REVIEWER	Nathalie Moise Columbia University Medical Center
REVIEW RETURNED	04-Jan-2019

GENERAL COMMENTS	This article assesses the effect of comorbid hypertension and MDD on initial CVD events. This remains an interesting study, and while the revisions helped clarify the findings, there remain a few concerns. The authors note that there is not a clear literature around the effect of MDD on first onset CVD (line 81), though there has been substantial literature (Nicholson notes that this depression is both a etiologic and prognostic risk factor for CVD, https://www.ncbi.nlm.nih.gov/pubmed/17082208). Do they mean no prior study has assessed comorbid MDD/hypertension on first CVD or lifetime MDD? The authors often answered several of my comments together, which often made it difficult to ascertain whether they had addressed every point. The definition of MDD is still not clear. The tracked version showed a paragraph that was added and then deleted. The authors note that this can be found in the supplementary material but this is such a key facet that it should be completely explained in the body of the paper (the authors can include the last sentence describing the MDD measure of the supplementary material in the body of the paper) The discussion focuses on possible reasons why MDD may be CVD protected and does not pay sufficient attention to their use of lifetime MDD as opposed to clinically significant current depressive symptoms that have been the focus on prior literature. Use of participants who are aware of and/or have sought treatment for MDD may also have complicated their findings.
---

REVIEWER	Christine Baumgartner Department of General Internal Medicine, Inselspital, Bern University Hospital, University of Bern, Switzerland
REVIEW RETURNED	05-Jan-2019

GENERAL COMMENTS	Major comment: Thank you for the revised version of the manuscript. While the results are interesting and confirm the results of previous studies showing that depression and hypertension are independently associated with incident cardiovascular events, the primary objective of the study is still not outlined clearly enough (is it the assessment of the independent relationship between MDD and CVD? Only among patients with hypertension, as stated in the abstract? Is it the assessment of interaction between MDD and hypertension on CVD?). In the introduction, the authors note the
---

	shortcomings of previous studies assessing interaction between MDD and hypertension on incident CVD to underline the novelty of this study, but their own assessment of multiplicative and additive interaction seems not to be presented in a very clear and understandable way, and lacks appropriate interpretation. I attached a paper from VanderWelle and Knol that could be helpful in more clearly describing the aim and interpreting interaction. Minor comments: In the introduction, the authors also make contradictory statements about the evidence on the association between MDD and incident cardiovascular events (page 6, line 83 “It is established that individuals with MDD are at increased risk of developing CVD”, while on line 94-95 MDD is well known to worsen post-cardiovascular event survival. The risk to first onset cardiovascular (?) is not known”) In the Methods on page 7, line 123, the age range of patients included in the study is not the same as the age range indicated in the abstract. Page 8, line 138: it is not clear how exclusions are “based on self report”. Please provide a reference from Smith et al. Page 9, line 163: I would prefer to see a more specific description of the assessment/measurement of MDD in the main body of the paper, and add that it was based on ICD-10 diagnostic criteria (as described in the abstract) Page 10, lines 186-189: I would suggest to move these numbers to the results section Page 13, line 269: what do the authors refer to when they state that “this” was noted to increase over time as a time-varying coefficient?
--	---

VERSION 2 – AUTHOR RESPONSE

Reviewer: 1

Reviewer Name: Nathalie Moise

Institution and Country: Columbia University Medical Center Please state any competing interests or state ‘None declared’: None declared

This article assesses the effect of comorbid hypertension and MDD on initial CVD events. This remains an interesting study, and while the revisions helped clarify the findings, there remain a few concerns.

The authors note that there is not a clear literature around the effect of MDD on first onset CVD (line 81), though there has been substantial literature (Nicholson notes that this depression is both a etiologic and prognostic risk factor for CVD, <https://www.ncbi.nlm.nih.gov/pubmed/17082208>). Do they mean no prior study has assessed comorbid MDD/hypertension on first CVD or lifetime MDD?

RESPONSE: Thank you for this point. Of note within Nicholson et al is that they state in their discussion that “incomplete and biased reporting of adjustment for conventional risk factors and the severity of coronary disease mean that these estimates for adjusted risk are likely to be inflated. Depression cannot, yet, be included in the group of established independent coronary risk factors.” We do agree that the assertion could be softened to state “The contribution on survival to first onset cardiovascular episode is less clear when MDD is stratified by hypertension” with reference to the above paper and we have additionally clarified the statement as follows “and no prior study has assessed comorbid MDD and hypertension on first episode cardiovascular disease.” (lines 81-83)

The authors often answered several of my comments together, which often made it difficult to ascertain whether they had addressed every point.

The definition of MDD is still not clear. The tracked version showed a paragraph that was added and then deleted. The authors note that this can be found in the supplementary material but this is such a key facet that it should be completely explained in the body of the paper (the authors can include the last sentence describing the MDD measure of the supplementary material in the body of the paper)

RESPONSE: The decision to remove the description was based on word count limitation. The authors are grateful for the reviewer’s suggestion to include the following description into the main text “Participants were classified as having a history of MDD if they reported at least one episode, which comprised of depression and/or irritability, with a duration of at least two weeks, plus had consulted with either a general practitioner or psychiatrist for mental ill-health. This classification followed the structured diagnostic approach within the International Classification of Diseases¹ and is described in more detail within the supplementary content.” (lines 146-150).

The discussion focuses on possible reasons why MDD may be CVD protected and does not pay sufficient attention to their use of lifetime MDD as opposed to clinically significant current depressive symptoms that have been the focus on prior literature. Use of participants who are aware of and/or have sought treatment for MDD may also have complicated their findings.

RESPONSE: We have expanded our discussion of the use of lifetime MDD rather than current depression scores to incorporate the inclusion of the debate surrounding the use of lifetime scores. We have included the statement “Lifetime MDD is thought to be under-reported in the literature. However, using current symptom scores may reduce power and precision because a smaller number of respondents would be identified as having an episode of MDD.² Given that we are assessing outcomes for which risk accumulates over a lifetime, we felt that a primary focus on lifetime episodes was appropriate.” (lines 317 – 321). Additionally, we have extended our limitations section to state “Participants who are aware of or had sought treatment for MDD may also have complicated our findings, however, our sensitivity analysis excluded those using pharmaceutical treatments and was in keeping with our main findings” (Lines 337 -339)

Reviewer: 2

Reviewer Name: Christine Baumgartner

Institution and Country: Department of General Internal Medicine, Inselspital, Bern University Hospital, University of Bern, Switzerland Please state any competing interests or state 'None declared': None

Please leave your comments for the authors below

Major comment:

Thank you for the revised version of the manuscript. While the results are interesting and confirm the results of previous studies showing that depression and hypertension are independently associated with incident cardiovascular events, the primary objective of the study is still not outlined clearly enough (is it the assessment of the independent relationship between MDD and CVD? Only among patients with hypertension, as stated in the abstract? Is it the assessment of interaction between MDD and hypertension on CVD?). In the introduction, the authors note the shortcomings of previous studies assessing interaction between MDD and hypertension on incident CVD to underline the novelty of this study, but their own assessment of multiplicative and additive interaction seems not to be presented in a very clear and understandable way, and lacks appropriate interpretation. I attached a paper from VanderWelle and Knol that could be helpful in more clearly describing the aim and interpreting interaction.

RESPONSE: We can clarify that the primary objective of this study was the assessment of the relationship between MDD and CVD in patients with hypertension. We have amended the statement regarding the shortcomings of previous studies in the introduction to say: "To date, survival analysis in comorbid hypertension and MDD have focussed on all-cause death³⁻⁵".(line 77 to 78) The paper's purpose is not to analyse the interaction effects per se but rather to identify that an interaction exists - a more detailed analysis of this issue will be the focus of a subsequent paper. We have now incorporated statistics from multiplicative and additive interaction in the time-stratified analysis. (sup table 11 and results section) In addition, we have clarified the use of the likelihood ratio test for assessing multiplicative interaction. " Presence of multiplicative interaction was calculated using the likelihood ratio test.⁶" (line 192-193)

Minor comments:

In the introduction, the authors also make contradictory statements about the evidence on the association between MDD and incident cardiovascular events (page 6, line 83 "It is established that individuals with MDD are at increased risk of developing CVD", while on line 94-95 MDD is well known to worsen post-cardiovascular event survival. The risk to first onset cardiovascular (?) is not known")

RESPONSE: We have rephrased the initial statement to say that "MDD is associated with CVD and worse long-term cardiovascular outcomes". (line 77) We also now say that: "The contribution on survival to first onset cardiovascular disease is less clear when MDD is stratified by hypertension and no prior study has assessed comorbid MDD and hypertension on first episode cardiovascular disease." (Line 80-82)

In the Methods on page 7, line 123, the age range of patients included in the study is not the same as the age range indicated in the abstract.

RESPONSE: The age range has been corrected to state 39 - 70

Page 8, line 138: it is not clear how exclusions are “based on self-report”. Please provide a reference from Smith et al.

RESPONSE: Exclusions were based on self-report inasmuch as those who listed schizophrenia or bipolar disorder from a list of pre-existing medical conditions given to UK Biobank, or fulfilled criteria for bipolar disorder as per Smith et al at baseline, were removed. Specific questions had been asked at baseline on angina, myocardial infarction (MI) or stroke also. A reference for Smith et al. has been added and the text has been amended to state “These exclusions were based on self-report (individuals who listed schizophrenia or bipolar disorder from a list of pre-existing medical conditions), or criteria for bipolar disorder as per Smith et al,¹ or where they responded “don’t know” or “prefer not to answer” to questions or data was missing that would limit our ability to exclude the presence of hypertension or MDD. Participants were further excluded from the adverse CVD outcome if they had a record of CVD prior to recruitment (self-reported angina, myocardial infarction (MI) or stroke based on specific questions, or previous hospital admission for angina, MI or stroke) (n= 9,172). For the stroke outcome this exclusion was limited to a record of stroke prior to baseline assessment (self-report or previous hospital admission for stroke) (n=2,280).” Lines 123 -131

Page 9, line 163: I would prefer to see a more specific description of the assessment/measurement of MDD in the main body of the paper, and add that it was based on ICD-10 diagnostic criteria (as described in the abstract)

RESPONSE: We have reintroduced the statement: “Participants were classified as having a history of MDD if they reported at least one episode, which comprised of depression and/or irritability, with a duration of at least two weeks, plus had consulted with either a general practitioner or psychiatrist for mental ill-health.” We have also added: “This classification followed the structured diagnostic approach within the International Classification of Diseases” (lines 146-150)

Page 10, lines 186-189: I would suggest to move these numbers to the results section

RESPONSE: This has been moved.

Page 13, line 269: what do the authors refer to when they state that “this” was noted to increase over time as a time-varying coefficient?

RESPONSE: We have clarified the statement to read “Although the MDD only HR was noted to increase over time as a time-varying coefficient.” (lines 240 -241)

1. Smith DJ, Nicholl BI, Cullen B, et al. Prevalence and characteristics of probable major depression and bipolar disorder within UK biobank: cross-sectional study of 172,751 participants. PLoS One 2013;8(11):e75362. doi: 10.1371/journal.pone.0075362
2. Patten SB. Accumulation of major depressive episodes over time in a prospective study indicates that retrospectively assessed lifetime prevalence estimates are too low. BMC psychiatry 2009;9:19-19. doi: 10.1186/1471-244X-9-19
3. Kuo PL, Pu C. The contribution of depression to mortality among elderly with self-reported hypertension: analysis using a national representative longitudinal survey. J Hypertens 2011;29(11):2084-90. doi: 10.1097/HJH.0b013e32834b59ad [published Online First: 2011/09/22]
4. Axon RN, Zhao Y, Egede LE. Association of depressive symptoms with all-cause and ischemic heart disease mortality in adults with self-reported hypertension. Am J Hypertens 2010;23(1):30-7. doi: 10.1038/ajh.2009.199
5. Hamer M, Batty GD, Stamatakis E, et al. The combined influence of hypertension and common mental disorder on all-cause and cardiovascular disease mortality. J Hypertens 2010;28(12):2401-6. doi: 10.1097/HJH.0b013e32833e9d7c
6. Marshall SW. Power for tests of interaction: effect of raising the Type I error rate. Epidemiologic Perspectives & Innovations 2007;4(1):4. doi: 10.1186/1742-5573-4-4

VERSION 3 – REVIEW

REVIEWER	Nathalie Niuse Columbia University Medical Center
REVIEW RETURNED	12-Apr-2019

GENERAL COMMENTS	The providers satisfactorily addressed my comments.
---

REVIEWER	Christine Baumgartner Inselspital, Bern University Hospital, University of Bern, Switzerland
REVIEW RETURNED	23-Apr-2019

GENERAL COMMENTS	My remaining concern is the confusing presentation of interaction (lines 289-294 and lines 302-304, and Supplementary Table 11). If both of the two exposures have an effect on the outcome (which is the case here according to this paper), then there must be interaction on some scale (VanderWeele & Knol, Epidemiol. Methods 2014; 3(1): 33–72). The description of the results on interaction should include whether interaction was found on the additive or multiplicative scale, with appropriate discussion.
---

VERSION 3 – AUTHOR RESPONSE

Reviewer: 1

Reviewer Name: Nathalie Niuse

Institution and Country: Columbia University Medical Center Please state any competing interests or state 'None declared': None

Please leave your comments for the authors below
The providers satisfactorily addressed my comments.

Authors response: The authors thank the reviewer for their comments.

Reviewer: 2

Reviewer Name: Christine Baumgartner

Institution and Country: Inselspital, Bern University Hospital, University of Bern, Switzerland Please state any competing interests or state 'None declared': None

My remaining concern is the confusing presentation of interaction (lines 289-294 and lines 302-304, and Supplementary Table 11). If both of the two exposures have an effect on the outcome (which is the case here according to this paper), then there must be interaction on some scale (VanderWeele & Knol, *Epidemiol. Methods* 2014; 3(1): 33–72). The description of the results on interaction should include whether interaction was found on the additive or multiplicative scale, with appropriate discussion.

Authors response: The authors thank the reviewer for their continued comments. While VanderWeele & Knol is a very useful and interesting paper, it does not discuss interaction with regards to proportional Hazard models and instead refers to other papers in this situation, (including Test for Additive Interaction in Proportional Hazards Models by Li and Chambless *Annals of Epidemiology* Volume 17, Issue 3, March 2007, Pages 227-236). This paper advocates the use of a likelihood ratio test in the situation of a multiplicative interaction in the situation of cox survival analysis, which has been done. Although this has been mentioned in the methods section. We have clarified which results are from the additive and multiplicative interaction within the results and discussion sections as well as adding further points to the discussion.

We have therefore edited the aforementioned lines as below:

“evidence of both additive and multiplicative interaction between hypertension and MDD at baseline for the overall cardiovascular outcome analysis before the 22.5 month time point (additive: RERI=0.563, 95%CI 0.189 - 0.938. Multiplicative: Likelihood ratio p-value 0.0116) and the female only cardiovascular endpoint analysis before the 29 month time point (additive: RERI=0.588, 95%CI 0.074 - 1.103. Multiplicative: Likelihood ratio p-value 0.031). However, after these time points there was no evidence of interaction on either the additive or multiplicative scale.”

Discussion

“In this large population cohort of middle-aged adults without CVD (adjusted for a broad range of confounders), individuals with co-morbid hypertension and MDD were at increased risk of CVD when compared to those with hypertension alone, MDD alone and neither condition. There was some evidence of additive and multiplicative interaction between hypertension and MDD at baseline, but not throughout follow-up and only within the female subgroup. Such a finding may suggest a causal interaction between MDD and hypertension in females only, but suggests that this may be limited over time leading to a suspected further interaction with a gender specific unmeasured confounder. Differences between co-morbid disease and either disease alone or no disease were more marked in females. For stroke outcomes, comorbid depression and hypertension was the only group that showed significantly increased HRs.”

In addition we have also stated in the possible mechanisms section in the discussion:

“Neuropeptide Y and oestrogen may represent a biologically plausible interaction between MDD and hypertension, however, this would require investigation.”

From supplementary table 11 we have eliminated the column for the Likelihood ratio chi statistic for clarity and instead left just the P-value, as well as highlighting which result is from the assessment and which is from the multiplicative interaction.

Supplementary Table 11: Relative excess risk due to interaction results on fully adjusted* models

Analysis	RERI	95% C.I.	LR test p-value
Adverse cardiovascular outcome before 22.5 months	0.563	(0.189 - 0.938)	0.0116
Adverse cardiovascular outcome after 22.5 months	-0.009	(-0.293 - 0.275)	0.563
Adverse cardiovascular outcome (males only)	0.058	(-0.240 - 0.357)	0.899
Adverse cardiovascular outcome (females only)before 29 months	0.588	(0.074 - 1.103)	0.031
Adverse cardiovascular outcome (females only)after 29 months	0.142	(-0.447 - 0.732)	0.5173
Stroke outcome	-0.047	(-0.485 - 0.391)	0.7271
Stroke outcome (males only)	-0.480	(-1.195 - 0.234)	0.1376
Stroke outcome (females only)	0.372	(-0.216 - 0.959)	0.314

*Adjusted for sociodemographic factors (age, sex, Townsend score, age of leaving full time education and ethnicity, history of diabetes, history of hypercholesterolemia, BMI, smoking history, alcohol use, systolic blood pressure, sedentary hours per day, physical activity and psychotropic medication use. RERI = Relative excess risk due to interaction (additive interaction), C.I.= Confidence interval, LR test = likelihood ratio test (multiplicative interaction)